# Exploring Foundation Model Adaptations for 3D Medical Imaging: Prompt-Based Segmentation with xLSTM network

Abdul Qayyum[1][0000−0003−3102−1595],  Moona Mazher [2][0000−0003−4444−5776],  and Steven A Niederer [1][0000−−0002−4612−6982]

[1] National Heart and Lung Institute, Imperial College London, UK
[2] Hawkes Institute, Department of Computer Science, University College London, London, UK
Email: a.qayyum@imperial.ac.uk

**Abstract.** Accurate segmentation of anatomical and pathological structures in 3D medical imaging is critical for effective diagnosis, treatment planning, and disease monitoring. Despite recent advances in deep learning, automated 3D medical image segmentation remains challenging due to anatomical variability, imaging artifacts, and the limited availability of annotated data. To address these issues, we present an interactive segmentation framework in the SAM-Med3D architecture with an xLSTM-UNet image encoder. Our encoder is specifically designed to capture long-range dependencies and hierarchical spatial features in volumetric medical data, improving contextual awareness while maintaining computational efficiency. We validate our approach using the CoreSet from the CVPR 2025 Foundation Models for 3D Biomedical Image Segmentation Challenge. Initial results demonstrate that our model achieves competitive performance in limited-scale testing, with DSC Final scores of 0.4855 (CT), 0.3071 (MRI), 0.4070 (PET), and 0.4458 (Ultrasound. NSD Final scores follow a similar trend, reaching 0.4992 (Ultrasound) and 0.4545 (CT). These early findings suggest strong potential for our architecture, particularly with further training on the full dataset. The proposed model supports multimodal prompts, including points and bounding boxes, allowing for flexible and intuitive user interaction a key requirement in clinical workflows. Our contributions include the development of a 3D-optimized interactive segmentation encoder, its integration into an existing foundation model framework, and an empirical evaluation that highlights the feasibility of our design. Future work will focus on full-scale training and refinement to bridge the performance gap with state-of-the-art methods.

**Keywords:** 3D medical image segmentation, Interactive segmentation, Vision transformers, xLSTM-UNet. SAM-Med3D, Foundation Models, User-Guided Refinement, Volumetric Data, Prompt-Based Segmentation.

# 1   Introduction

Medical image segmentation plays a critical role in various clinical tasks, including diagnosis, treatment planning, and disease monitoring. Accurate delineation of anatomical structures or pathological regions in 3D medical scans is essential for reliable clinical decision-making. However, automated segmentation remains a challenging problem due to the high variability in anatomical shapes, ambiguous boundaries, imaging artifacts, and the scarcity of annotated data. Furthermore, medical data is often acquired in 3D volumes, which makes the segmentation process more computationally intensive and complex compared to natural 2D images.

Recent advances in deep learning, especially vision transformers and convolutional neural networks (CNNs), have significantly improved segmentation accuracy. Yet, these models typically require large-scale annotated datasets, which are expensive and time-consuming to obtain in the medical domain. Additionally, these models lack flexibility in adapting to new tasks without retraining, highlighting the need for interactive and generalizable segmentation frameworks. In the domain of natural image segmentation, foundation models such as the Segment Anything Model (SAM) [6]and its improved variant SAM2 [11] have demonstrated impressive zero-shot and few-shot capabilities. These models leverage powerful vision-language pretraining to generalize across image domains and tasks. However, directly applying these models to medical images has limitations due to domain differences and the lack of medical-specific priorities.

To address this, adaptations like MedSAM [8] and MedSAM2 [10] were proposed to better fit the medical image domain. These models incorporated domain-specific fine-tuning and data augmentations. Despite improvements, they still suffer from limited interactive refinement ability and lack support for multimodal prompts, such as textual descriptions or radiological terms, which are common in medical diagnostics. To further enhance interaction, methods such as SegVol [1], SAM-Med3D [12], VISTA3D [3], and nnInteractive [2], have introduced mechanisms for user-guided refinement, such as point-based corrections or region proposals.

Despite recent progress, a significant gap remains in developing interactive medical image segmentation frameworks that are both robust and efficient for 3D volumetric data. Our work is motivated by the need to enhance interactive segmentation while addressing domain-specific challenges in medical imaging. We build upon the SAM-Med3D framework, known for its prompt-based interaction capabilities, and propose architectural enhancement through a proposed encoder design.

Specifically, we introduce a customized encoder based on xLSTM-UNet, a hybrid architecture aimed at capturing long-range dependencies and hierarchical features within 3D medical volumes. This encoder is integrated

into the SAM-Med3D pipeline, where we retain the original prompt decoder to support user-driven segmentation via points and bounding boxes. The design aims to balance contextual awareness with computational efficiency, tailored for medical scenarios.

Our main contributions are as follows:
1.    We propose a novel xLSTM-UNet encoder, optimized for 3D medical imaging tasks, and integrate it within the interactive SAM-Med3D framework. Our model supports a variety of prompt types, enhancing usability in clinical settings through flexible user interaction.
2.    We conduct initial experiments using the CVPR 2025: Foundation Models for 3D Biomedical Image Segmentation Challenge CoreSet dataset, training our encoder from scratch to evaluate its design independently.
3.    While our proposed encoder does not yet outperform strong baselines like VISTA3D or nnInteractive, it demonstrates promising results in limited-scale testing.

Due to hardware and time constraints, we were unable to train our proposed model on the full challenge dataset. As a result, the performance reported here reflects partial training on the CoreSet. We plan to extend our evaluation to the full dataset in future work to better assess the potential of our approach and its capacity to close the performance gap with state-of-the-art methods.

## 2    Method

Our proposed model is shown in Figure.1. The detail of each component is described in the following sections.

### 2.1    Proposed xLSTM encoder

The xLSTM block is a hybrid architectural component designed to integrate convolutional feature extraction with sequential modeling capabilities provided by a modified Long Short-Term Memory (mLSTM) unit. This integration is especially beneficial for medical image analysis, where both local spatial patterns and long-range dependencies across slices are crucial. The xLSTM begins with a convolutional layer that extracts spatial features from the input slice. This is followed by instance normalization (IN) to stabilize training and Leaky ReLU activation to introduce non-linearity, which together enhance the learning of meaningful spatial representations.

After this initial processing, the feature map is flattened and normalized before being divided into two distinct processing pathways. In the first pathway, the features undergo a linear transformation followed by the SiLU activation function, allowing for complex nonlinear interactions within local regions. The second pathway introduces a flip mechanism, which reverses the input along the slice axis to enable bidirectional processing. This flipped sequence is passed through the mLSTM, which captures long-range dependencies by modeling relationships between distant slices. The outputs of both pathways are then

merged to form a unified representation.

To ensure effective information flow and model stability, the merged features are further refined through a final linear transformation and combined with the original input via a residual connection. This residual link preserves input information and supports efficient gradient propagation during training. The inclusion of the flip operation is key to capturing both past and future contextual information, providing a bidirectional perspective that is vital for tasks such as segmentation where anatomical continuity across slices is important.

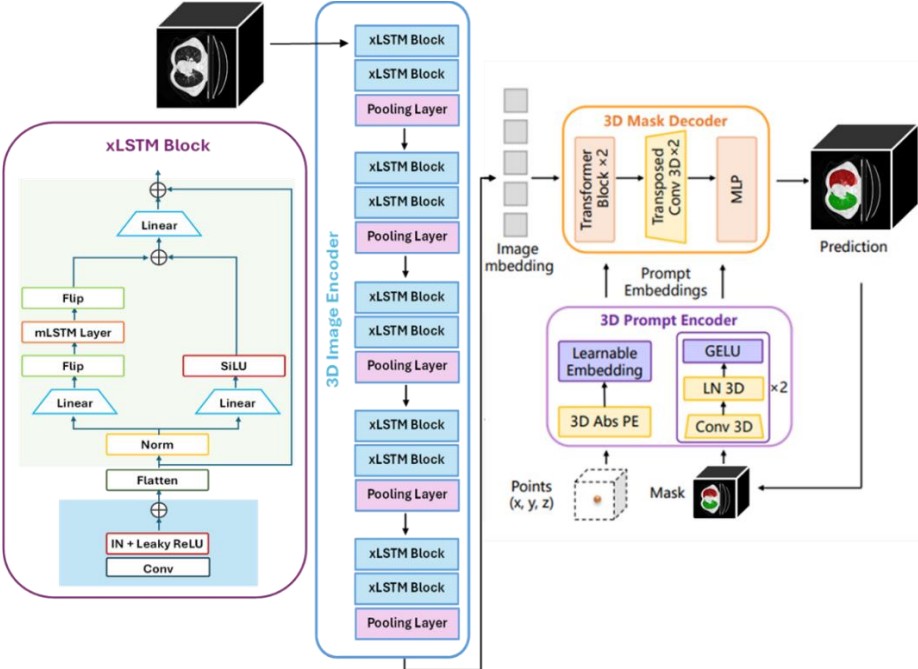

**Fig. 1.** Overview of the proposed interactive segmentation framework. The architecture integrates a customized image encoder based on xLSTM-UNet, a prompt encoder for handling interactive inputs such as points and bounding boxes, and a mask decoder that generates the final segmentation output. This modular design enables effective processing of 3D medical volumes while supporting flexible user interaction for refined and accurate segmentation.

In our proposed encoder module, the xLSTM is employed across five encoder layers, each also incorporating convolutional and pooling operations. This multi-layer design allows the model to hierarchically extract features while modeling slice-to-slice relationships in 3D medical images. Unlike Vision Transformers (ViTs), which rely on self-attention mechanisms, our xLSTM-based approach offers a more efficient and tailored method for volumetric medical data. By combining convolutional and recurrent components, the xLSTM ensures both local detail preservation and global contextual awareness, which are essential for accurate and coherent segmentation outcomes.

## 2.2   3D Prompt Encoder

The 3D prompt encoder in our architecture is responsible for embedding user-provided interaction cues—such as point coordinates, bounding boxes, or coarse masks—into a latent representation that guides the mask generation process. This module begins by converting raw prompt inputs into dense learnable embeddings. Each interaction type (e.g., positive point, negative point, or mask) is assigned a unique token that is further enriched through positional encoding. Specifically, we use 3D absolute positional encoding (3D Abs PE) to ensure that the spatial location of the prompts within the volumetric space is preserved. These embeddings are passed through a series of lightweight but expressive 3D convolutional layers to model local spatial interactions, followed by non-linear activation (GELU) and layer normalization (LayerNorm) to ensure stable feature transformation. By encoding spatial and semantic context from the user prompts, the 3D prompt encoder effectively translates interactive inputs into a form that the mask decoder can leverage to produce anatomically meaningful segmentation results. Importantly, this module remains unchanged from the SAM-Med3D pipeline to preserve its proven interactive refinement capabilities.

## 2.3   3D Mask Decoder

The 3D mask decoder is a critical component of the segmentation pipeline, tasked with synthesizing both the image features (from the encoder) and the prompt embeddings (from the prompt encoder) into a coherent segmentation output. This component operates on the fused feature space and is composed of a stack of Transformer blocks adapted for 3D inputs. These blocks allow for both self-attention (within image features) and cross-attention (between image and prompt features), enabling the model to contextualize anatomical structures relative to the spatial hints provided by the user. After attention-based fusion, the decoder incorporates several 3D convolutional layers that refine the transformer outputs by capturing fine-grained spatial patterns. These layers are followed by normalization and activation functions to improve gradient flow and feature discriminability. Finally, the decoder ends with a multi-layer perceptron (MLP) head that maps the refined features to voxel-level predictions. The output is a 3D binary or multi-class mask, depending on the target task. By maintaining the original design of the 3D mask decoder from SAM-Med3D, we ensure strong support for interactive segmentation while integrating our custom encoder module seamlessly. In our experiments, the prompt encoder and 3D mask decoder were frozen, while the xLSTM-UNet encoder was fine-tuned on the training dataset.

## 3   Experiments

### 3.1   Dataset and evaluation metrics

The development set is an expanded version of the dataset introduced in the

CVPR 2024 MedSAM on Laptop Challenge [9], including more 3D cases from public datasets (A complete list is available at https://medsam-datasetlist.github.io/) and covering commonly used 3D modalities, such as Computed Tomography (CT), Magnetic Resonance Imaging (MRI), Positron Emission Tomography (PET), Ultrasound, and Microscopy images. The hidden testing set is created by a community effort where all the cases are unpublished. The annotations are either provided by the data contributors or annotated by the challenge organizer with 3D Slicer [5] and MedSAM2 [10]. In addition to using all training cases, the challenge contains a coreset track, where participants can select 10% of the total training cases for model development. For each iterative segmentation, the evaluation metrics include Dice Similarity Coefficient (DSC) and Normalized Surface Distance (NSD) to evaluate the segmentation region overlap and boundary distance, respectively. The final ranking metrics are defined as follows:

- DSC_AUC and NSD_AUC: Area Under the Curve (AUC) scores for DSC and NSD, representing cumulative segmentation improvement over five iterative user interactions. The AUC is computed only for click-based predictions, excluding the optional initial bounding box.
- Final DSC and NSD: Scores measured after all refinement steps, reflecting the final segmentation accuracy.
- Algorithm runtime is restricted to 90 seconds per class; exceeding this limit results in all DSC and NSD metrics for that case being set to zero.

### 3.2   Implementation details

**Preprocessing** Following the practice in MedSAM [8], all images were processed to npz format with an intensity range of $[0, 255]$. Specifically, for CT images, we initially normalized the Hounsfield units using typical window width and level values: soft tissues (W:400, L:40), lung (W:1500, L:-160), brain (W:80, L:40), and bone (W:1800, L:400). Subsequently, the intensity values were rescaled to the range of $[0, 255]$. For other images, we clipped the intensity values to the range between the 0.5th and 99.5th percentiles before rescaling them to the range of $[0, 255]$. If the original intensity range is already in $[0, 255]$, no preprocessing was applied. We have cropped a 3D volume in spatial size 128x128x128 and resampled the whole dataset for training the model.

**Environment settings** The development of environments and requirements are presented in Table 1. The training protocol used in our proposed model is shown in Table 2.

**Table 1.** Development environments and requirements.

| | |
|---|---|
| **System** | Ubuntu 20.04.6 LTS |
| **CPU** | AMD Ryzen Threadripper PRO 5955WX (16 cores / 32 threads) |
| **RAM** | 128 GB (16 × 8 GB), 2.67 MT/s |
| **GPU** | NVIDIA RTX A6000 48 GB |
| **CUDA version** | 12.2 |
| **Programming language** | Python 3.8.3 |
| **Deep learning framework** | PyTorch 2.0.1+cu118, Torchvision 0.15.2+cu118 |

**Table 2.** Training protocol.

| Configuration | Details |
|---|---|
| **Pre-trained Model** | None |
| **Batch Size** | 12 |
| **Patch Size** | 128 × 128 × 128 |
| **Total Epochs** | 200 |
| **Optimizer** | Adam |
| **Initial Learning Rate (lr)** | $8 \times 10^{-4}$ |
| **LR Decay Schedule** | MultiStepLR (milestones: [120, 180], gamma=0.1) |
| **Gradient Accumulation Steps** | 20 |
| **Weight Decay** | 0.1 |
| **Loss Function** | DiceCELoss |
| **Number of Model Parameters** | 24,867,472 |
| **Training Time** | 6 days |
| **Image Size** | 128 |

**Training protocols**

In our experiments, we adopted the same training protocols described in SAM2 [11] to ensure consistency and fair comparison. The following outlines the key components of the training procedure:

1.  Data Augmentation:
To improve generalization across diverse medical imaging modalities and reduce overfitting, extensive 3D data augmentation strategies were applied. These include random cropping, flipping, rotation, intensity scaling, and Gaussian noise injection. Such augmentations help the model learn robust representations under varying anatomical and imaging conditions.

2.  Data Sampling Strategy:
We employed a patch-based sampling strategy, where volumetric image patches were sampled around annotated structures or user-provided prompts. This focused sampling allows the model to better learn localized anatomical features and improves efficiency during training, especially for high-resolution 3D volumes.

3.  Model Selection Criteria:
The optimal model was selected based on validation performance, using the average Dice Similarity Coefficient (DSC) and Normalized Surface Dice (NSD) scores as the primary metrics. The model achieving the highest average DSC on the CoreSet validation split was selected for evaluation and comparison.

## 4   Results and discussion

### 4.1   Quantitative results on validation set

Table. 3 shows the performance evaluation of our proposed xLSTM encoder within the SAM-Med3D framework was conducted across five imaging modalities CT, MRI, microscopy, PET, and ultrasound—using standard metrics: DSC AUC, NSD AUC, DSC Final, and NSD Final. Overall, our model demonstrated promising trends in terms of foundational performance, but it underperformed compared to state-of-the-art models such as VISTA3D and nnInteractive. For CT images, the proposed model achieved a DSC Final of 0.4855, which, while lower than VISTA3D (0.8041) and nnInteractive (0.8764), is competitive given that our model was trained on a limited dataset and from scratch. In MRI, a similar trend was observed, with our model achieving 0.3071 DSC Final versus 0.7302 for nnInteractive, highlighting room for improvement in capturing complex anatomical structures. For microscopy, the model did not achieve any segmentation accuracy (DSC = 0), likely due to modality-specific domain shifts and insufficient adaptation, suggesting the need for modality-specific fine-tuning.

In PET imaging, the model achieved a moderate DSC Final of 0.4070, outperforming the baseline SAM-Med3D (0.5344) in terms of NSD AUC in some settings, though trailing behind more mature methods. Lastly, for ultrasound, the model yielded a DSC Final of 0.4458, which is an improvement over SegVol (0.3109), but still below nnInteractive (0.8547). These results reflect the current limitations of our partially trained model but also demonstrate its potential as a generalizable encoder. With full training and further optimization, particularly through integration into mature frameworks like VISTA3D or 3DSegVol, we expect significant performance gains across all modalities.

**Table 3.** Quantitative evaluation results of the validation set on the **coreset track.**

| Modality | Methods | DSC AUC | NSD AUC | DSC Final | NSD Final |
|---|---|---|---|---|---|
| CT | SAM-Med3D | 2.2408 | 2.2213 | 0.5590 | 0.5558 |
| | VISTA3D | 3.1689 | 3.2652 | 0.8041 | 0.8344 |
| | SegVol | 2.9809 | 3.1235 | 0.7452 | 0.7809 |
| | nnInteractive | 3.4337 | 3.5743 | 0.8764 | 0.9165 |
| | Proposed model | 1.9326 | 1.8080 | 0.4855 | 0.4545 |
| MRI | SAM-Med3D | 1.5222 | 1.5226 | 0.3903 | 0.3964 |
| | VISTA3D | 2.5895 | 2.9683 | 0.6545 | 0.7493 |
| | SegVol | 2.6719 | 3.1535 | 0.6680 | 0.7884 |
| | nnInteractive | 2.6975 | 3.0292 | 0.7302 | 0.8227 |
| | Proposed model | 1.1861 | 1.1291 | 0.3071 | 0.3115 |
| Microscopy | SAM-Med3D | 0.1163 | 0 | 0.0291 | 0 |
| | VISTA3D | 2.1196 | 3.2259 | 0.5478 | 0.8243 |
| | SegVol | 1.6846 | 2.9716 | 0.4211 | 0.7429 |
| | nnInteractive | 2.3311 | 3.1109 | 0.5943 | 0.7890 |
| | Proposed model | 0 | 0 | 0 | 0 |
| PET | SAM-Med3D | 2.1304 | 1.7250 | 0.5344 | 0.4560 |
| | VISTA3D | 2.6398 | 2.3998 | 0.6779 | 0.6227 |
| | SegVol | 2.9683 | 2.8563 | 0.7421 | 0.7141 |
| | nnInteractive | 3.1877 | 3.0722 | 0.8156 | 0.7915 |
| | Proposed model | 1.6267 | 0.9885 | 0.4070 | 0.2473 |
| Ultrasound | SAM-Med3D | 1.4347 | 1.9176 | 0.4102 | 0.5435 |
| | VISTA3D | 2.8655 | 2.8441 | 0.8105 | 0.8079 |
| | SegVol | 1.2438 | 1.8045 | 0.3109 | 0.4511 |
| | nnInteractive | 3.3481 | 3.3236 | 0.8547 | 0.8494 |
| | Proposed model | 1.6074 | 1.8157 | 0.4458 | 0.4992 |

## 4.2   Qualitative results on validation set

We have validated our proposed model using representative examples of both good and poor segmentation performance across different modalities, as

illustrated in Fig. 2 and Fig. 3. Fig. 2 presents successful segmentation results for CT, MR, PET, and ultrasound (US) images, where the proposed model accurately delineates anatomical structures with high overlap compared to ground truth, as shown in Column 3. These examples highlight the model's ability to capture relevant features and boundaries effectively in varied imaging contexts. In contrast, Fig. 3 showcases failure cases where the model struggles to produce accurate segmentations, particularly in regions with ambiguous boundaries, low contrast, or structural variability. These visualizations underscore the strengths of our model in common clinical scenarios while also revealing limitations that motivate further refinement.

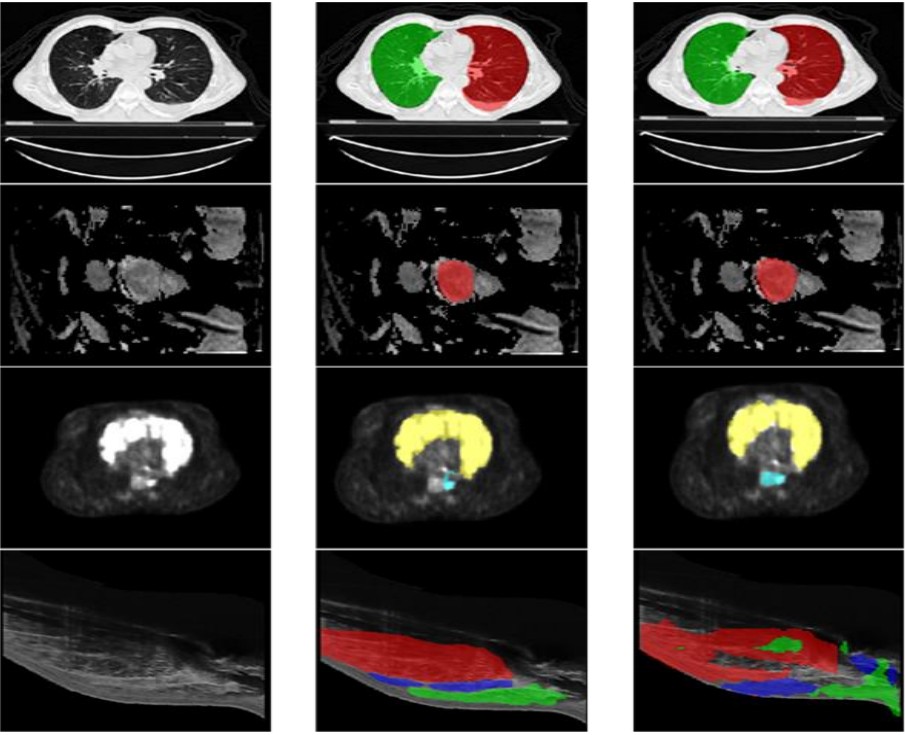

**Fig. 2.** Representative examples of the best segmentation performance achieved by the proposed model across all evaluated modalities. The first column displays the original input images, the second column overlays the ground truth annotations on the input, and the third column overlays the model's predicted segmentations. These cases demonstrate the model's ability to accurately capture anatomical and pathological structures, showing high spatial agreement with the ground truth even in complex 3D medical imaging scenarios.

### 4.3  Limitations and future work

While our proposed xLSTM-UNet encoder integrated into the SAM-Med3D framework shows promising early results, several limitations remain. Due to hardware and time constraints, our current experiments were limited to partial training on the CVPR 2025 Foundation Models for 3D Biomedical Image

Segmentation Challenge CoreSet. As a result, the performance does not yet fully reflect the model's capacity. Furthermore, although the xLSTM-UNet design is intended to capture long-range spatial dependencies in 3D volumes, it currently underperforms compared to leading interactive methods, likely due to limited training scale and lack of comprehensive hyperparameter tuning. The framework presently supports only basic prompts (points and bounding boxes), lacking richer or multimodal guidance (e.g., textual or anatomical region descriptors) that could improve usability in clinical settings.

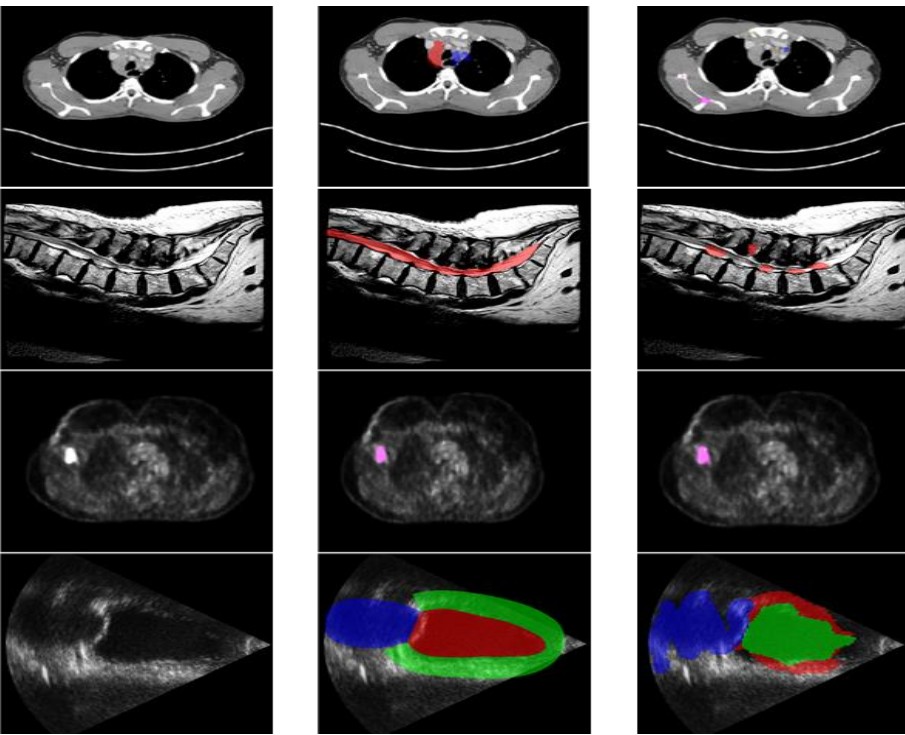

**Fig. 3.** Poor segmentation examples across all evaluated modalities using the proposed model. The first column displays the input images, the second column shows the ground truth annotations overlaid on the input, and the third column presents the model's predicted segmentations overlaid on the same input images. These examples illustrate the model's limitations in challenging scenarios such as low-contrast regions, irregular anatomical structures, or noisy inputs, where segmentation quality significantly degrades.

Additionally, the xLSTM layers introduce moderate computational overhead, which may affect scalability for real-time or low-resource environments. In future work, we plan to train our model on the full challenge dataset, explore optimizations for inference efficiency, and extend support for multimodal and domain-aware prompting. Importantly, we also intend to evaluate the generalizability of our xLSTM-UNet encoder by integrating it into other

advanced foundation model frameworks such as VISTA3D and 3DSegVol, enabling a broader and more rigorous assessment of its effectiveness across segmentation paradigms.

## 5    Conclusion

In this work, we proposed an xLSTM-UNet encoder designed to enhance interactive 3D medical image segmentation within the SAM-Med3D framework. Our motivation stemmed from the limitations of existing foundation models in handling the unique challenges of 3D biomedical data, such as long-range spatial dependencies, ambiguous anatomical boundaries, and limited annotated datasets. By integrating the xLSTM-UNet architecture, combining convolutional hierarchies with LSTM-style sequence modeling into the SAM-Med3D pipeline, we aimed to improve both contextual representation and user-guided segmentation performance.

Initial experiments were conducted using a subset of the CVPR 2025 Foundation Models for the 3D Biomedical Image Segmentation Challenge CoreSet. Despite the limited training scale, our model demonstrated promising segmentation accuracy and effective prompt-based interaction capabilities. While the xLSTM-UNet did not yet surpass state-of-the-art methods like VISTA3D or nnInteractive in quantitative performance, it showed competitive results and practical feasibility, especially in scenarios with minimal supervision.

Our findings suggest that incorporating long-range sequence modeling into 3D encoders is a viable direction for enhancing foundation model frameworks in medical imaging. Future work will involve training on the full challenge dataset, supporting multimodal prompts, and testing the encoder's adaptability in other interactive frameworks such as VISTA3D and 3DSegVol. Ultimately, this research contributes toward building more flexible, interactive, and generalizable foundation models for 3D biomedical image segmentation

**Acknowledgements**  We thank all the data owners for making the medical images publicly available and CodaLab [13] for hosting the challenge platform.

**Disclosure of Interests.**  The authors have no competing interests to declare that are relevant to the content of this article.

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

**Table 7.** Checklist Table. Please fill out this checklist table in the answer column. (**Delete this Table in the camera-ready submission**)

| Requirements | Answer |
| --- | --- |
| A meaningful title | Yes |
| The number of authors (≤6) | 3 |
| Author affiliations and ORCID | Yes |
| Corresponding author email is presented | Yes |
| Validation scores are presented in the abstract | Yes |
| Introduction includes at least three parts: background, related work, and motivation | Yes |
| A pipeline/network figure is provided | 1 |
| Pre-processing | 6 |
| Strategies to data augmentation | 6 |
| Strategies to improve model inference | 6 |
| Post-processing | 6 |
| Environment setting table is provided | 1 |
| Training protocol table is provided | 2 |
| Ablation study | Nill |
| Efficiency evaluation results are provided | 3 |
| Visualized segmentation example is provided | 2 and 3 |
| Limitation and future work are presented | Yes |
| Reference format is consistent. | Yes |
| Main text >= 8 pages (not include references and appendix) | Yes |