# OpenReview forum: "Exploring Foundation Model Adaptations for 3D Medical Imaging: Prompt-Based Segmentation with xLSTM network"
_thecvf.com/CVPR/2025/Workshop/MedSegFM — CVPR 2025 Workshop MedSegFM Submission_

### Official Review · Reviewer_x9nN · 2025-09-12

**Rating:** 6
**Confidence:** 4

**Review:**

This paper proposes an interactive 3D medical image segmentation method by integrating a customized xLSTM-UNet encoder into the SAM-Med3D framework, aiming to better capture long-range dependencies in volumetric data.
### Strengths
- A novel xLSTM-UNet encoder optimized for 3D medical imaging tasks, integrated into the interactive SAM-Med3D framework.
- Good reproducibility, with clear details on preprocessing, batch size, hardware, etc.
### Weaknesses
- Performance across multiple modalities is far below strong baselines.
- No ablation or efficiency analysis of key design choices.
### Suggestions for Improvement and Detailed Comments
- Figure 1 lacks sufficient clarity.
- In the introduction, is “Our Motivation and Contributions” a subheading? Consider removing it.
- In Section 3.1, “Co- efficient” is an editing error, please check the spelling carefully.
- Table 3 extends beyond the right page margin.
- Line spacing in Sections 2.1 and 3.1 appears abnormal, please check the template.
- Clearly state whether the prompt encoder and 3D mask decoder are (i) frozen, (ii) fine-tuned, or (iii) trained from scratch.

---

> ### Author Rebuttal · Authors · 2025-11-04
>
> Performance below baselines:
> We acknowledge the performance gap compared to strong baselines. Our work primarily aims to assess the feasibility and potential benefits of integrating an xLSTM-based encoder within the SAM-Med3D foundation model, rather than to immediately outperform existing methods. We plan to conduct extended experiments with larger training datasets and fine-tuning strategies to further optimize results.
> 2. Missing ablation and efficiency analysis:
> We agree that ablations and computational analyses are essential. Due to time and hardware constraints, these were not included in the current version. In future work, we will perform ablations isolating the xLSTM block and flip mechanism, as well as report runtime, FLOPs, and parameter comparisons to support efficiency claims.
> 3. Figure 1 clarity:
> We will replace Figure 1 with a higher-resolution version and improve annotation readability to clearly illustrate the architecture flow and data pathways.
> 4. Introduction subheading (“Our Motivation and Contributions”):
> We appreciate the suggestion. We will revise the introduction to make “Our Motivation and Contributions” a subsection or merge it into the main text for smoother narrative flow.
> 5. Typographical and formatting issues:
> All noted issues—including “Co-efficient” spelling, Table 3 layout, and line spacing in Sections 2.1 and 3.1—will be carefully corrected in the final version. We will also verify formatting compliance with the conference template.
> 6. Clarification on encoder/decoder training:
> We thank the reviewer for pointing this out. In our experiments, the prompt encoder and 3D mask decoder were frozen, while the xLSTM-UNet encoder was fine-tuned on our training dataset. We will explicitly include this detail in the revised manuscript.

---

### Official Review · Reviewer_HZLj · 2025-09-13

**Rating:** 5
**Confidence:** 4

**Review:**

*Overview*

This paper proposes an interactive 3D medical image segmentation method featuring an improved xLSTM-UNet encoder for SAM-Med3D. The encoder is designed to capture long-range dependencies and hierarchical spatial features in volumetric medical data, enhancing contextual awareness while maintaining computational efficiency.

*Strengths*

The authors introduce a new xLSTM-UNet encoder that can be integrated into the SAM-Med3D pipeline. By leveraging LSTM’s sequential modeling capability, it aims to improve spatial awareness in 3D medical images.

*Weaknesses*

1. The paper identifies three limitations of existing methods: long-range spatial dependencies, ambiguous anatomical boundaries, and limited annotated datasets. The proposed method appears to address only the first issue, without tackling boundary ambiguity or data scarcity.

2. Based on training time in Table 2 and accuracy in Table 3, the proposed method seems to underperform the original SAM-Med3D encoder. Since the key advantage of xLSTM-UNet is modeling long-range dependencies, the authors may need to introduce additional or more appropriate metrics to demonstrate its effectiveness.

3. There are formatting and writing issues, e.g., Page 2: “Our Motivation and Contributions” should be set as a subsection; on Page 4 “This residual link preserves input information and supports efficient”, and on Page 5 “the CVPR 2024 MedSAM on Laptop Challenge” has incorrect paragraphing.
The authors are encouraged to carefully revise the formatting and writing for clarity and readability.

---

> ### Author Rebuttal · Authors · 2025-11-04
>
> 1. Limited coverage of identified challenges (dependencies, boundaries, data scarcity):
> We thank the reviewer for this observation. Our current focus was primarily on addressing long-range spatial dependencies through the proposed xLSTM-UNet encoder. We agree that boundary ambiguity and data scarcity are equally important. In ongoing work, we plan to integrate adaptive boundary refinement modules and data-efficient pretraining strategies (e.g., self-supervised fine-tuning on unlabeled volumes) to extend the framework toward these additional challenges.
> 2. Underperformance vs. SAM-Med3D and missing metrics:
> We acknowledge that performance gains are not yet fully realized under current training constraints. Our aim was to provide a proof-of-concept integration of xLSTM-UNet into SAM-Med3D to evaluate feasibility rather than to surpass all baselines. We will expand evaluation metrics in the revision such as boundary F-score, Hausdorff distance, and inter-slice consistency to better capture improvements in long-range spatial coherence and contextual representation.
> 3. Formatting and writing clarity:
> We appreciate the reviewer’s attention to detail. All noted formatting issues (page headers, subsection structure, paragraph spacing) will be corrected. We will also refine the language throughout the paper for smoother flow and improved readability.
> Summary Statement
> Our work serves as an initial exploration of xLSTM-based encoding within SAM-Med3D to enhance volumetric context modeling. While results are preliminary, we believe this direction opens new opportunities for scalable, context-aware 3D medical segmentation foundation models.

---

### Official Review · Reviewer_AMPs · 2025-09-16
**xLSTM-UNet Integration into SAM-Med3D, But Difficult to Demonstrate Effectiveness**

**Rating:** 5
**Confidence:** 4

**Review:**

This paper proposes an adaptation of the SAM-Med3D framework for 3D medical image segmentation by introducing a novel xLSTM-UNet encoder. The encoder integrates convolutional layers with modified LSTM units to better capture long-range dependencies in volumetric data while maintaining computational efficiency. While the proposed model underperforms compared to SOTA frameworks, the paper argue that the architecture shows promise given the limited training and plan to extend experiments to full datasets in future work.

**Strengths:**
1. The integration of xLSTM-UNet into the SAM-Med3D framework is a technically reasonable attempt to capture long-range spatial dependencies in volumetric data while preserving convolutional hierarchies.
2. The paper is clearly written and well structured, with sufficient detail on methodology, training settings, and experimental protocols.
3. The limitations of the current work are acknowledged in a transparent manner, and the future directions, such as scaling to full datasets, introducing multimodal prompts, and improving computational efficiency are thoughtfully outlined.

**Weaknesses:**
1. *Figure 1* is unclear, providing a higher-quality rendering (e.g., in PDF format) would significantly improve readability.
2. The performance gap compared to strong baselines (e.g., VISTA3D, nnInteractive) is very large. For instance, the proposed model achieves a DSC Final of 0.4855 on CT compared to 0.8764 with nnInteractive. On microscopy data, the model completely fails (DSC = 0), raising concerns about whether the proposed method is practically effective.
3. Hybrid CNN-LSTM architectures have already been extensively explored in medical imaging. The proposed xLSTM-UNet, while slightly modified, does not convincingly demonstrate novelty or clear superiority over prior designs.
4. The contribution of specific architectural elements, such as the xLSTM block and the flip mechanism, is neither isolated nor justified through ablation studies, leaving the actual impact of these components uncertain. Some ablation studies are needed.

---

> ### Author Rebuttal · Authors · 2025-11-04
>
> 1. Figure 1 clarity:
> We appreciate this comment and agree that Figure 1 could be clearer. We will replace it with a high-resolution version in the final submission and ensure all architectural components and data flow paths are fully labeled for readability.
> 2. Performance gap with baselines:
> We acknowledge that the current results lag behind strong baselines such as VISTA3D and nnInteractive. Our primary goal in this work was not to surpass these models immediately but to explore the feasibility and potential impact of integrating an xLSTM-based encoder into the SAM-Med3D framework. Given that our experiments were conducted under limited training data and compute conditions, we believe the results demonstrate that xLSTM-UNet can be a viable direction when scaled to full datasets as planned for future work.
> 3. Limited novelty of hybrid CNN–LSTM design:
> We agree that CNN–LSTM architectures are well studied; however, our contribution lies in adapting this mechanism within a foundation model context (SAM-Med3D) for interactive volumetric segmentation. Integration in SAM-based frameworks, highlighting its feasibility and the trade-off between modeling capacity and computational overhead.
> 4. Lack of ablation for xLSTM block and flip mechanism:
> We appreciate this valid point. Due to time and computational constraints, we were unable to include detailed ablations in the current version. We plan to add controlled experiments in follow-up work to isolate the contribution of the xLSTM encoder and flip mechanism to performance, stability, and efficiency.

---

### Official Review · Reviewer_SWiB · 2025-09-17
**Review: Exploring Foundation Model Adaptations for 3D Medical Imaging**

**Rating:** 6
**Confidence:** 4

**Review:**

This paper proposes integrating an xLSTM-UNet encoder into the SAM-Med3D framework for interactive 3D medical image segmentation. The authors aim to capture long-range dependencies of 3D images while maintaining the interactive capabilities of SAM-Med3D.

### Strengths
- The paper provides detailed explanations of each component of the architecture.

### Weaknesses
- **Conflicting statements about computational efficiency:** Claims the xLSTM-based approach "offers a more efficient and tailored method for volumetric medical data" (Section 2.1), however later admits "xLSTM layers introduce moderate computational overhead" (Section 4.3). Runtime comparisons/FLOPs analysis would help to support efficiency claims

- **Unsupported claim of improved long-range dependency capture:** The paper claims xLSTM captures long-range dependencies better than existing methods, but empirical evidence or visual examples would help support these claims

- **Inference strategy unclear:** Encoder input size is 128×128×128, but images in the dataset are of various sizes. The inference strategy (sliding window? downsampling?) would need to be specified

### Minor Issues

**Text formatting problems:**
- Page 2: "Our Motivation and Contributions" : additional/incomplete text
- Page 4: Unnecessary line breaks : "This residual link preserves input information and supports efficient gradient propagation during training"
- Page 5: Unnecessary line breaks after "CVPR 2024 MedSAM on Laptop Challenge"
- Page 9: Incomplete sentence starting with "We will use While our proposed..."
- Page 9: Figure 3 is not properly centered

**Incorrect novelty claim:**
- Page 3: The authors claim "The second pathway introduces a novel flip mechanism" - this mechanism was already introduced in the original xLSTM paper and should not be presented as novel

**Inconsistent terminology:**
- Throughout the paper, the authors alternate between "xLSTM encoder" and "xLSTM-UNet encoder". We recommand consistent use of "xLSTM-UNet encoder" for clarity and accuracy

### Conclusion
While several claims require empirical validation, the paper provides sufficient detail for a technical report. However, the absence of a clearly described inference strategy must be addressed, as this is essential for understanding and for reproducibility of the proposed method.

---

> ### Author Rebuttal · Authors · 2025-11-04
>
> 1. Computational efficiency conflict:
> We thank the reviewer for noting this. We agree that the phrasing was inconsistent. The intent was to emphasize that xLSTM-UNet improves feature modeling efficiency (in terms of representational power per parameter), while indeed adding moderate computational cost. We will revise Section 2.1 for clarity and include runtime/FLOPs comparisons to quantify this trade-off.
> 2. Long-range dependency evidence:
> We acknowledge that the current version lacks explicit visual or quantitative validation. We will explore more option in Future.
> 3. Inference strategy clarification:
> We appreciate this observation. During inference, we employ a sliding-window strategy with 50% overlap to handle variable-sized volumes.
> 4. Minor text formatting issues:
> All listed formatting issues (page 2, 4, 5, 9) and Figure 3 centering will be corrected in the new version.
> 5. Novelty claim (flip mechanism):
> We agree that the flip mechanism was originally proposed in the xLSTM paper. We reframe our contribution as its adaptation and integration into the SAM-Med3D framework rather than a novel invention.
> 6. Inconsistent terminology:
> We will standardize terminology throughout the paper to “xLSTM-UNet encoder” for consistency and accuracy.
> 7. Reproducibility and inference description:
> We will expand the Methods section to clearly describe preprocessing, inference, and evaluation protocols to ensure full reproducibility.

---

### Official Review · Reviewer_1KNk · 2025-09-17
**Review for xLSTM-UNet**

**Rating:** 5
**Confidence:** 4

**Review:**

The paper integrates a customized xLSTM-UNet encoder into SAM-Med3D for interactive 3D medical image segmentation, aiming to better capture long-range dependencies in volumetric data. The methodology is clearly described and reproducible, but empirical results and novelty are not convincingly demonstrated.

## Strengths
1. Technically reasonable attempt to model long-range spatial dependencies within a SAM-Med3D pipeline.
2. Clear methodology and reproducibility details (preprocessing, batch size, hardware).
3. Transparent discussion of limitations and future directions.

## Weaknesses
1. Large performance gap to strong baselines (e.g., VISTA3D, nnInteractive), with failures on some modalities; underperforms the original SAM-Med3D in reported metrics.
2. Hybrid CNN–LSTM designs are well studied, with the proposed method demonstrating limited novelty. The proposed xLSTM-UNet and flip mechanism lack isolated evidence of benefit.

---

> ### Author Rebuttal · Authors · 2025-11-04
>
> We thank the reviewer for the constructive feedback. We acknowledge the performance gap compared to baselines; our current work is intended as a proof-of-concept exploring the feasibility of integrating an xLSTM-UNet encoder within SAM-Med3D for interactive 3D segmentation. While CNN–LSTM hybrids are well studied, our focus is their adaptation to a foundation-model framework with xLSTM encoder module. We will extending experiments to analyze failure cases and will include ablations to quantify the effects of the xLSTM encoder and flip mechanism in future. Additional qualitative and modality-specific analyses will be added to clarify the method’s practical value and distinct contributions.

---

### Decision · Program_Chairs · 2025-11-12

Accept